# Internet of Medical Things Healthcare for Sustainable Smart Cities: Current Status and Future Prospects

**Priyanka Mishra** * **and Ghanshyam Singh** *

Centre for Smart Information and Communication Systems, University of Johannesburg, Johannesburg 2006, South Africa
* Correspondence: priyankam@uj.ac.za (P.M.); ghanshyams@uj.ac.za (G.S.)

**Abstract:** The concept of smart and connected healthcare has emerged in response to the growing demand for the improvement of healthcare systems and the increasing prevalence of chronic diseases. Looking towards the future, smart healthcare holds great potential to transform the healthcare industry by providing more efficient, personalized, and accessible healthcare services. This paper delves into the concept of intelligent, interconnected, and customized healthcare systems within the Internet of Medical Things (IoMT) framework. It explores the utilization of cutting-edge technologies, including the IoMT, in conjunction with big data, cloud computing, artificial intelligence, and blockchain to provide healthcare services that are not only more efficient but also more convenient and personalized. It draws on existing literature, bibliometric data, and global marketing analysis to gain a deeper understanding of these technologies and their impact on the healthcare system. We have explored several upcoming features of the Healthcare 5.0 paradigm, which represents the next evolution in healthcare systems focusing on a more personalized and patient-centric approach. We introduce a healthcare architecture specifically designed for the IoMT that prioritizes the security considerations associated with devices. Finally, we have focused on addressing open research challenges, particularly those related to fundamental social needs, such as ensuring equitable access to smart and connected healthcare systems.

**Keywords:** sustainable smart cities; smart and connected healthcare; internet of things; internet of medical things; Healthcare 4.0/5.0

## 1. Introduction

Due to the significant population growth, traditional healthcare is unable to meet the needs of everyone. Accessing medical services is not affordable or accessible for everyone, despite excellent infrastructure and cutting-edge technologies. Smart healthcare emerges as a response to the challenges faced by traditional healthcare systems, which are constrained by limited resources and a growing demand for services. It aims to make healthcare more intelligent, efficient, and sustainable by leveraging advanced technologies and innovative approaches [1]. By providing enhanced access to healthcare, smart healthcare can enhance overall citizen health, alleviate the strain on existing healthcare infrastructure, and contribute to the advancement of smarter and more sustainable cities. The technology facilitates remote monitoring of patients and reduces the cost of treatment. Additionally, it allows medical practitioners to expand their services across regional boundaries. In the context of the growing trend toward smart cities, a smart healthcare system ensures that its citizens live a healthy life. The introduction of the Internet of Things (IoT) into the healthcare system has brought forth the concept of the Internet of Medical Things (IoMT), which has redefined smart and connected healthcare systems (healthcare 4.0/5.0) globally. It has the potential to reduce healthcare costs by enabling more accurate diagnoses and treatments and reducing the need for costly hospital visits. It leverages emerging technologies such as electronic health records, telemedicine, and health information technology.

This system utilizes sensors to gather data, transmits data through the IoT, and processes data via cloud computing [2]. Through the coordination and integration of social systems, smart healthcare enables the dynamic and refined management of human society. It is designed to provide individuals with personalized and effective healthcare services while improving cost efficiency in healthcare delivery. It has the capacity to enhance the quality of life by granting better access to medical information and services. Wearable devices, the IoT, and mobile internet technologies facilitate dynamic access to information, connecting individuals, materials, and healthcare institutions within an active and responsive medical ecosystem. Overall, the concept of smart healthcare signifies an advanced stage in the development of medical technology [3]. It promotes interaction among all stakeholders in the healthcare field, ensuring the provision of necessary services, facilitating informed decision-making, and optimizing resource allocation within the realm of smart healthcare. However, the apprehension of security threats and risks, particularly in the field of medicine, poses significant challenges due to the sensitivity of data and critical information in IoMT-based systems.

*Related Work*

In recent times, significant attention has been devoted to developing healthcare services and addressing various technical and architectural challenges in the healthcare sector. Researchers and professionals have actively engaged in designing and implementing healthcare solutions to enhance patient care, improve healthcare delivery processes, and foster better communication and collaboration among healthcare providers. Various technologies have been introduced by researchers and developers, providing new approaches for creating more efficient, effective, and secure healthcare platforms. These technologies, based on existing literature with methods and purposes in the healthcare domain, are shown in Table 1. For instance, in [4], artificial intelligence (AI) and machine learning (ML) solutions were utilized to enhance COVID-19 analysis and screening, enabling faster diagnosis and customized treatment and control measures. In [5], deep learning (DL) was employed to successfully screen COVID-19 patients using X-radiography (X-rays) and Computed Tomography (CT) scans with 100% accuracy. In [6], Layered Recurrent Neural Networks (L-RNN) are used to predict missing data in Hepatocellular Carcinoma data. In [7], several ML algorithms are compared with detect brain tumors from magnetic resonance (MR) images, with Partial Tree (PART) outperforming other techniques. AI and ML techniques for monitoring cardiovascular diseases highlight the challenge of interpreting black box models [8]. Furthermore, Ref. [9] applies deep reinforcement learning and neural networks (NN) to optimize spectrum access. The articles emphasize the need for big data analytics to address issues such as sustainable information and communication technology (ICT) [10,11]. Additionally, AI/ML techniques are employed for security and privacy in networks [12], proposing an AI-driven hybrid framework for intrusion detection in IoT-enabled e-health. Authors in [13–17] discuss the use of blockchain and cryptography in the e-healthcare domain to enhance security and privacy. For contact tracing in crowded areas while maintaining COVID patients' privacy, a proposed physical unclonable function (PUF)-based host tracking system is introduced [18]. PUF-based sensors secure physical measurements [19] and utilize PUF-based sensor devices for secure monitoring of COVID-19 patients [20]. The use of Software Defined Networking (SDN) in the IoMT for network management and slicing is exemplified in [21–24]. These works showcase the potential of SDN in IoMT for efficient network management and improved service provisioning. Additionally, IoMT applications addressing COVID-19 challenges are presented in [25], while Ref. [26] discusses the challenges of data-intensive ecosystems in cyber-physical systems (CPS), particularly in the context of mobile healthcare environmental monitoring. A wireless sensor network (WSN)-based system has been designed to track the heart rate and movement of elderly individuals within their residences. The system can send alerts to healthcare professionals, caregivers, or family members via a smartphone if there are significant changes in physiological indicators such as falls, high

heart rate, or low heart rate [27]. Researchers are actively working on enhancing healthcare systems through advanced technologies such as AI and IoT [28,29]. Several studies have also explored the integration of various computing technologies, including IoT and cloud computing, to develop smart healthcare solutions [30]. These solutions hold the potential to tackle healthcare issues and improve patient care by leveraging the capabilities of these advanced technologies. For instance, a smart healthcare framework based on cloud and edge computing has been proposed [31], which can be implemented in smart cities for progressive developments. Similarly, a framework has been developed for individuals with voice disorders using edge computing and cloud frameworks [32]. The aforementioned studies emphasize the potential benefits of integrating different domains to develop more efficient and effective healthcare systems. This paper stands out by offering a comprehensive review of key literature concepts while highlighting the implementation of a robust IoMT architecture using practical services. Moreover, it effectively demonstrates how these technologies can be practically applied in smart and connected healthcare settings. This integration of theoretical and practical aspects of IoMT makes it a valuable addition to the field. The author's contributions in this context are highlighted below:

- The exploration of smart and connected healthcare systems enabled by the IoMT has encompassed various dimensions, leveraging existing literature, bibliometric data, and global marketing analysis. In addition, an examination of the upcoming features of the Healthcare 5.0 paradigm has been conducted.
- A comprehensive overview of the most promising enabling technologies for smart and connected healthcare systems has been provided, including IoMT, big data analytics, blockchain, healthcare cloud, fog, and edge computing. Additionally, the development requirements for these technologies have been elucidated.
- The presentation of a robust and generalized healthcare architecture for the IoMT, taking into account security constraints for the devices, has been accomplished. Furthermore, promising practical applications/services that can benefit from this architecture have been highlighted.
- Despite considerable investments in research and development, healthcare systems still face numerous challenges. These challenges have been discussed in relation to fundamental social needs, privacy, security vulnerabilities, regulations, and rights. Additionally, potential areas of research have been outlined to address these challenges and realize the vision of smart and connected healthcare.

**Table 1.** Technologies based on existing literature in the healthcare domain.

| Technology | References | Methods Used | Purposes |
|---|---|---|---|
| Artificial Intelligence/Machine Learning | [4] | ML and AI Algorithms | Radio Imaging Technology, CT Scan, X-ray, and Blood Sample Data |
| | [5] | DL Methods | X-rays and CT scans |
| | [6] | L-RNN | To predict the missing data in the Hepatocellular Carcinoma data |
| | [7] | ML algorithms: (Partial Tree (PART), Random Forest, Naive Bayes, and Random Tree) | To detect brain tumor from the MR images |
| | [8] | Several AI/ML methods | Monitor Cardiovascular Diseases |
| | [9] | Deep Reinforcement Learning and Neural Networks methods | Improvement in latency, error rate, etc. |
| | [10,11] | Big Data Analytics | Overcoming issues such as green and sustainable ICT |
| | [12] | AI/ML method | To detect DDoS and some privacy attacks |

**Table 1.** *Cont.*

| Technology | References | Methods Used | Purposes |
|---|---|---|---|
| Blockchain | [13] | IoMT assisted Blockchain | Stress management |
| | [14] | Blockchain based authentication method | Decentralization, reliability and security for medical devices |
| | [15,16] | Blockchain technology | Secure management of EHR (Electronic Health Record), EMR (Electronic Medical Records), and PHR (Personal Health Records) |
| Cryptography | [17] | Symmetric and Asymmetric Cryptography methods | Security requirements of medical data |
| Physical Unclonable Function (PUF) | [18] | PUF based host tracking system | Tracing in the crowded area taking into account the privacy of COVID patients |
| | [19] | PUF based sensors | Secure physical measurements |
| | [20] | PUF based sensor devices | Securely monitor for COVID-19 patients |
| SDN/IoMT | [21] | SDN orchestration | Combat cyber threats |
| | [22] | IoMT based cyber training framework | Orthopedic surgery using next generation internet technology |
| | [23] | Non-Orthogonal Multiple Access scheduling method | Improvement in energy consumption, network delay, and effective throughput |
| | [24] | IoT enabled e-healthcare management system | Traffic management |
| | [25] | IoMT technologies merged with AI, Big data, and blockchain | Technology and security management in COVID-19 situation |
| CPS | [26] | Big data systems | Mobile healthcare environmental monitoring and security vulnerabilities |
| WSN | [27] | Wireless sensor network-based intelligent system for public health | Home care monitoring systems |
| Computing Technology | [28] | DL | Wearable device for diagnosis and combating CIVID-19 |
| | [29] | Embedded NN Techniques | Smart mobiles devices for better computing |
| | [30] | Cloud Computing | Smart health solutions |
| | [31] | Edge Computing | Smart healthcare framework in smart cities |
| | [32] | Edge computing with cloud framework | Voice disorder treatments |

This paper is unique in its approach as it not only provides a comprehensive review of the key concepts and attributes of literature, but also implements a robust architecture using practical and existing services of the IoMT. Additionally, it demonstrates the practical applicability of these technologies in the context of smart and connected healthcare. This is an innovative contribution to the field as it combines both theoretical and practical aspects of IoMT in healthcare.

The remaining sections of this paper are organized as follows, as depicted in Figure 1. Section 2 provides an explanation of Smart and Connected Healthcare. Section 3 discusses the Bibliometric/Global Market Analysis, while Section 4 focuses on the evolution of healthcare. In Section 5, the Internet of Medical Things is explored, followed by Section 6, which delves into various enabling technologies in sustainable smart cities. Section 7 discusses the IoMT healthcare services and applications. Section 8 highlights the open research challenges and research directions. Finally, Section 9 concludes the paper.

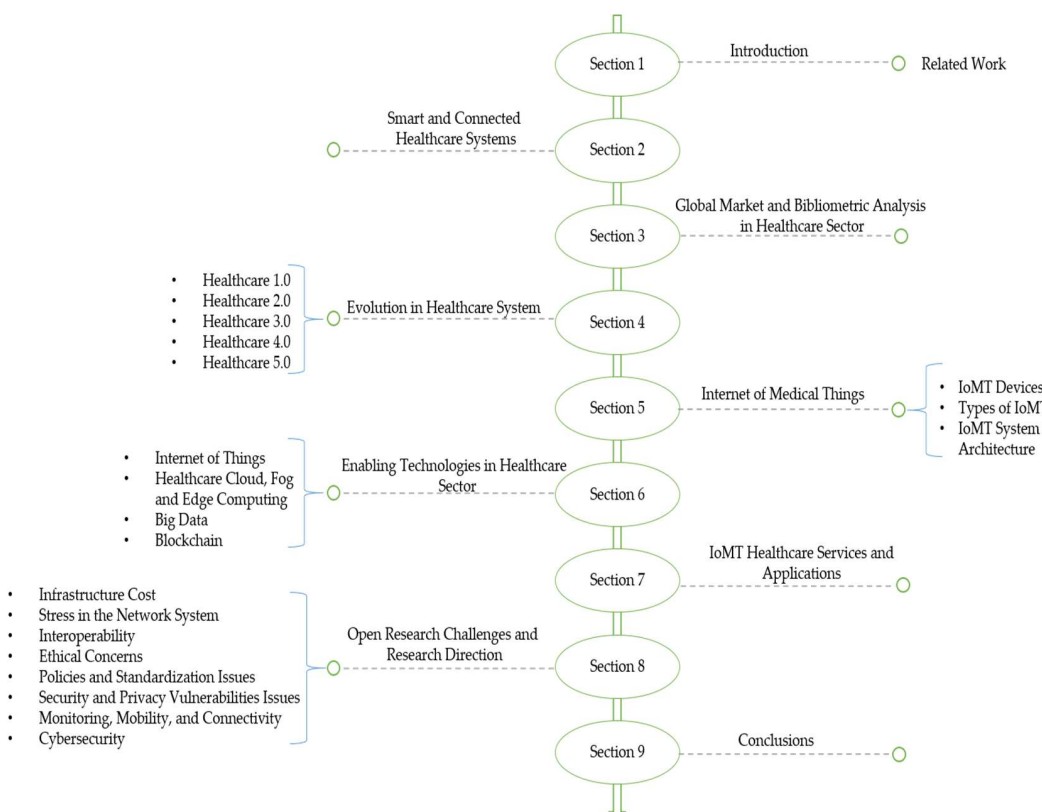

**Figure 1.** Organization of this Research Paper.

## 2. Smart and Connected Healthcare Systems

The term "connected health" encompasses digital healthcare solutions that enable remote monitoring, encompassing continuous health monitoring, emergency response, and alarm functionality. The primary objectives of connected health are to enhance the quality and efficiency of healthcare by enabling self-care and supplementing it with remote care. This concept originated in telemedicine, wherein users receive timely feedback regarding their health status as required. A smart healthcare solution operates autonomously, while a connected healthcare approach offers clinical feedback to users. The smart and connected healthcare sector is classified based on how it is consumed by end users, which determines the economy of the industry. Figure 2 illustrates the broad classification of this sector, taking into account services, medical devices, technologies used, applications, system management, and end users. The integration of connectivity technologies enables the expansion of applications within the healthcare system. Wireless technologies facilitate the integration of small devices, enabling remote health monitoring via the IoT [33]. However, it is important to maintain constant internet connectivity and support heavy data traffic in a hospital setting where a healthcare network is managed using Wi-Fi and cables.

Smart and connected health refers to digital healthcare solutions and systems that are fully connected and accessible remotely [34]. The National Science Foundation (NSF) and the National Institutes of Health (NIH) initiated this concept in 2013 to promote the development and integration of innovative information technology approaches in healthcare [35]. The research team aimed to foster multidisciplinary collaboration and generate innovative "smart" ideas to enhance scientific collaborations in the field. Technology, particularly artificial intelligence, has played a crucial role in advancing healthcare delivery within the framework of smart cities. Smart and connected health research involves various disciplines, including medical informatics, public health, big data, bioengineering, and the telecommunications industry. In complex healthcare scenarios, smart and connected health requires resource-aware, time-constrained, complex, and secure healthcare transactions involving multiple stakeholders. Furthermore, smart and connected health has the poten-

tial to revolutionize healthcare by accelerating treatment and testing procedures, reducing physician visits, efficiently responding to emergencies and pandemics, and enhancing the quality of patient care [36].

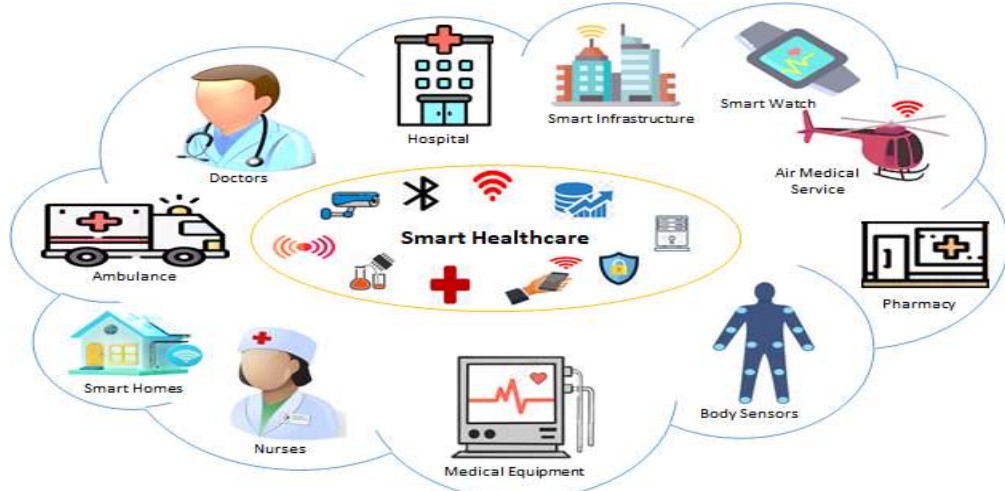

**Figure 2.** Smart and Connected Healthcare System Concept.

## 3. Global Market and Bibliometric Analysis in the Healthcare Sector

The global IoMT healthcare market has experienced significant growth, primarily driven by the increased adoption of technologies in the healthcare sector and rapid technological advancements. This connectivity enables medical devices to be connected to the internet, facilitating enhanced healthcare services. Government initiatives and private investments in digital technology have also contributed to the growth of the healthcare sector. As a result, the global IoMT market is expected to benefit from these factors. It is projected that the IoMT market will generate over US$ 172.4 billion by 2030, with a compound annual growth rate (CAGR) of 15.9% from 2021 to 2030 [37], as illustrated in Figure 3. Furthermore, key market players are anticipated to adopt new growth strategies, creating new opportunities for expansion. The ongoing pandemic has prompted rapid changes in the healthcare industry, and there is a likelihood of significant transformations in the entire healthcare sector in the future, with a focus on addressing pandemic situations.

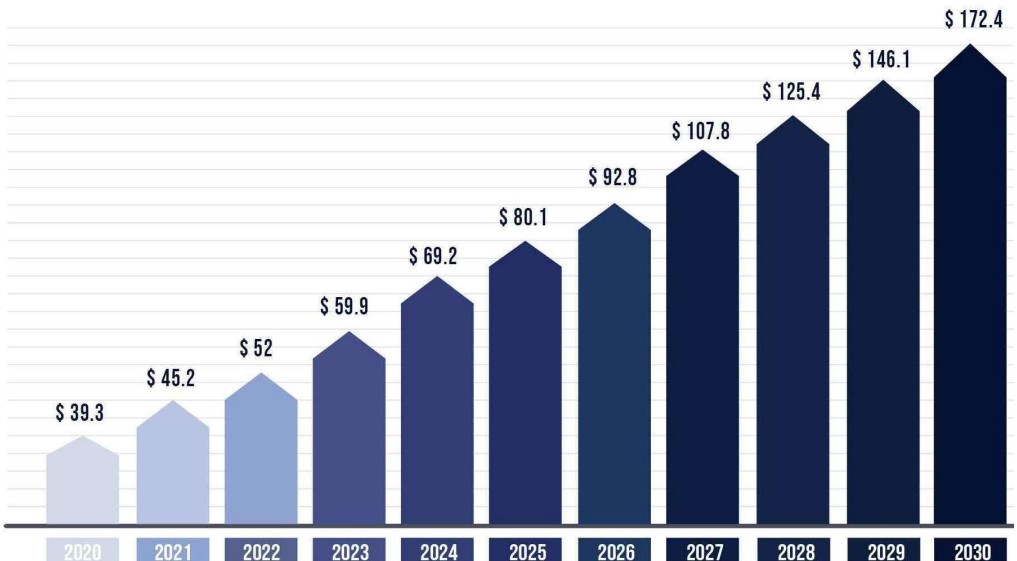

**Figure 3.** Internet of Things market size, 2020 to 2030 (USD Billion) [37].

The bibliometric analysis focused on IoMT healthcare in smart cities and utilized data from the Scopus database. The analysis primarily focused on articles as the primary type of publication in this domain. VOS viewer software was employed to conduct visualization analysis, which aided in identifying research gaps in the field of IoMT for smart cities. This bibliometric analysis provides valuable insights for researchers seeking to explore unexplored areas within the IoMT and smart cities domains. In this data analysis, Figure 4a illustrates the number of documents published per year and the number of publications in healthcare systems. The most publications will be made in the year 2022 with 523, and as publications are growing each year, this is an excellent field for further research. Figure 4b shows the documents per year by source. IEEE Access has more than 60 publications. Figure 4c shows the congregation of keywords that occur together in a paper. The interrelations between keywords that co-appear in the research are linked together. Internet of Things, healthcare, blockchain, big data, wearable sensors, and smartphones are the most common keywords.

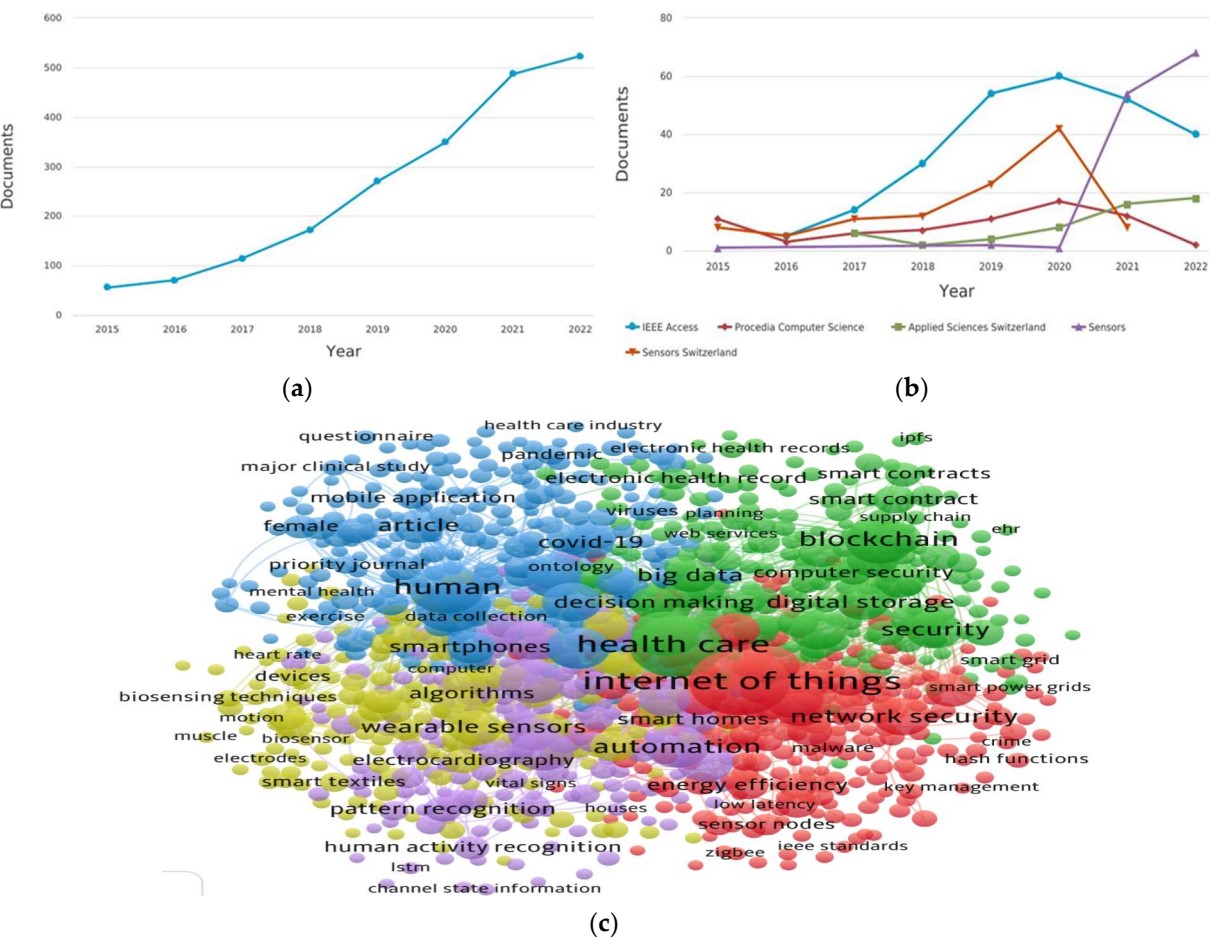

**Figure 4.** The published documents per year: (**a**) with the number of publications; (**b**) source; (**c**) the congregation of co-appearing keywords from the Scopus database.

Overall, the field of IoMT in healthcare for sustainable smart cities is rapidly growing and attracting researchers at various levels of expertise. The potential of IoMT for addressing hazardous situations and improving work efficiency is highly promising. The increasing number of publications in this field indicates a significant research gap that can be explored further. Smart healthcare systems are continually integrating new technologies to enhance precision and suitability in urban settings. However, it is crucial to assess both the benefits and limitations of these technologies to leverage their advantages and find solutions for any drawbacks. This ongoing cycle of evaluation and improvement

allows us to progress in this direction and discover new technologies while addressing their limitations.

## 4. Evolution in Healthcare

The evolution of Healthcare from 1.0 to 5.0 represents the transformative journey of the healthcare industry, driven by advancements in technology and patient-centric care. This evolution represents a journey towards a future where technology seamlessly integrates into all aspects of healthcare, empowering patients, enhancing outcomes, and driving greater efficiency and sustainability in the healthcare domain. Healthcare delivery has undergone a series of evolutions and revolutions over time, represented by multiple stages from Healthcare 1.0 to Healthcare 5.0, as depicted in Figure 5.

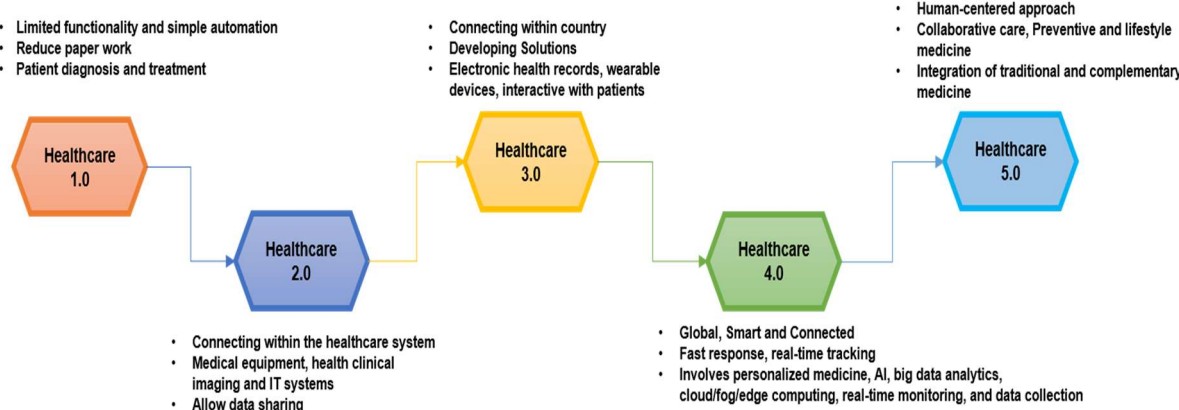

**Figure 5.** The potential development of the healthcare 1.0 to healthcare 5.0.

### 4.1. Healthcare 1.0

It refers to the basic encounter between a patient and a physician. Patients visit clinics and meet with physicians and other healthcare practitioners during such visits. Clinicians perform consultations, tests, diagnoses, and prescribe medications, as well as follow-up plans (e.g., ordering lab tests or imaging tests, referring patients to specialists). Hundreds of years have passed since this model was first used in healthcare [38].

### 4.2. Healthcare 2.0

The concept of Healthcare 2.0 emphasizes the use of technology and data to improve patient care. Rather than treating symptoms only, it takes a holistic approach to healthcare, which includes technologies such as MRI, ultrasound, CT scanning, pulse oximeters, and arterial lines [38]. A variety of technologies are being used, such as electronic health records, telemedicine, and big data analytics. In order to provide personalized care, it collects, stores, and analyzes data from patients using digital tools. Besides monitoring patients remotely, it can provide healthcare providers with real-time feedback. Moreover, it facilitates collaboration between healthcare providers and can improve outcomes.

### 4.3. Healthcare 3.0

In addition to being more efficient and cost-effective, this type of healthcare delivery is also more convenient for patients and providers. Furthermore, it enables improved communication between patients and providers, as well as enhanced access to care for individuals residing in rural areas. Moreover, it allows for more accurate, better treatment, and improved patient outcomes [38]. The availability of computer networks has made remote care and tele-health possible, and electronic visits (such as communicating with a physician through a patient portal) are beginning to replace some face-to-face encounters. With the current COVID-19 pandemic, telehealth and virtual visits have become more popular.

### 4.4. Healthcare 4.0

Healthcare 4.0 primarily focuses on the integration and utilization of advanced technologies such as AI, big data analytics, and computing technologies in healthcare systems [39]. It emphasizes the digitalization of healthcare processes, the connectivity between different healthcare stakeholders and systems, and the promotion of personalized medicine through the analysis of patient data and genomics. Efficiency and automation are key aspects of Healthcare 4.0, enabling streamlined processes and improved healthcare delivery. Additionally, it empowers patients by providing them access to their health data and involving them in decision-making processes.

### 4.5. Healthcare 5.0

Healthcare 5.0 takes a more human-centered approach, placing a strong emphasis on holistic well-being and patient engagement [40]. It promotes collaborative partnerships among healthcare professionals, patients, families, and communities to co-create care plans and make shared decisions. Healthcare 5.0 recognizes the impact of social determinants of health, such as socio-economic factors and lifestyle, and addresses them as integral components of healthcare. Preventive measures, health promotion, and lifestyle interventions play a significant role in Healthcare 5.0, shifting the focus from solely managing diseases to maintaining overall health. Moreover, Healthcare 5.0 integrates traditional and complementary medicine alongside modern medicine, acknowledging the value of diverse healing practices. The ultimate goal of Healthcare 5.0 is to achieve better health outcomes for patients while also reducing costs and improving overall healthcare quality [41].

## 5. Internet of Medical Things

Wireless communication technology, driven by advancements in connectivity and mobility, has become an essential enabler in the digital era. This technology facilitates the seamless transmission of data and has gained significant attention from major carriers around the world. As wireless communication continues to evolve, the IoT emerges as a prominent paradigm, revolutionizing the way devices, objects, and systems interact and exchange information. With its ability to connect and interconnect countless smart devices, the IoT is poised to shape the future of networks, ushering in a new era of interconnectedness and intelligent communication. Initially, the concept of IoT was primarily focused on business and industry, but it has now expanded its scope to include homes and offices. IoT has become an integral component of modern living, and it is expected to be a crucial element of the future 6G network [42–44]. By 2025, healthcare is projected to account for one third of IoT devices, contributing approximately 40% of the total value of IoT technology [45].

The term "m-health" is used to describe healthcare services that utilize mobile computing, medical sensors, and communication technologies. This emerging field has the potential to greatly improve access to healthcare for both patients and providers. By leveraging mobile devices, m-health enables the efficient and cost-effective delivery of healthcare services remotely. Telemedicine, telemonitoring, and patient education are examples of how m-health is utilized to provide healthcare services beyond traditional clinical settings. With m-health, healthcare professionals are able to remotely monitor patients, diagnose and treat them, and offer them information and support. This approach not only enhances convenience but also has the potential to improve the quality of care, reduce healthcare costs, and increase healthcare accessibility for underserved populations. The integration of 6LowPAN with evolving 4G networks enables the development of future internet-based m-health services, which are referred to as m-IoT. It is important to note that m-IoT exhibits certain characteristics inherent to the global mobility of participants. This gives rise to the conceptualization of mobile IoT services. In a study [46], the use of m-IoT architecture and implementation was investigated for noninvasively measuring glucose levels. Wireless Sensor Networks, Wireless Body Area Networks, and Radio Frequency Identification (RFID) allow monitoring of patient position and status [47–50]. It is possible

to track chronic diseases with these wearable devices. Wearables powered by artificial intelligence can track steps, heart rate, blood pressure, calories, and more. The Wearable Internet of Things (WIoT) [51], the Internet of Health Things (IoHT) [52], the Internet of Medical Things (IoMT) [53], the Internet of Nano Things (IoNT) [54], and the Internet of Mobile Health Things (m-IoT) [46] are some of the variants derived from the core IoT concept in the healthcare fields depicted in Figure 6.

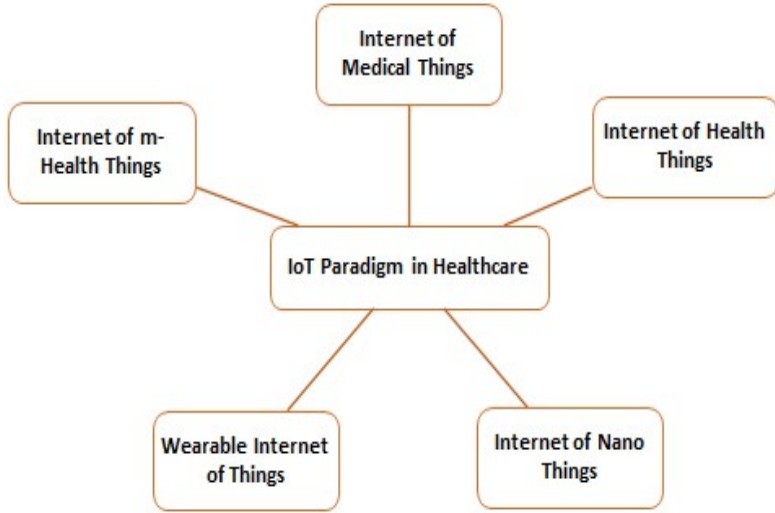

**Figure 6.** IoT concept in Healthcare Domain.

### 5.1. IoMT Devices

The IoMT is a network of medical devices, sensors, and other healthcare-related equipment connected to the internet to collect, process, and transmit data for healthcare purposes. It refers to a wide range of connected medical devices and applications that collect and transmit patient data for analysis and monitoring. There are several types of IoMT devices, including:

- Wearables: These medical devices, including smartwatches, fitness trackers, and health monitors, are worn on the body and track various health parameters such as heart rate, blood pressure, and sleep patterns.
- Implantable: These medical devices, such as pacemakers and insulin pumps, are implanted inside the body to provide continuous monitoring and treatment for chronic conditions.
- Remote monitoring devices: These medical devices, such as blood glucose monitors and blood pressure monitors, remotely monitor patients and transmit real-time data to healthcare professionals for informed decision-making in patient care.
- Telemedicine applications: These applications, such as video conferencing and messaging apps, enable remote communication between patients and healthcare professionals, particularly benefiting those in remote or underserved areas.
- Medical imaging devices: These devices, such as X-rays, CT scans, and MRIs, capture medical images that are transmitted to healthcare professionals for analysis and diagnosis.
- Smart hospital equipment: Connected medical devices used in hospital settings, such as patient monitoring systems and infusion pumps, provide real-time data on patient health, empowering healthcare professionals to make well-informed decisions regarding patient care.

Here is a classification in Table 2 for IoMT, categorized by the type of devices and their application areas, with examples. These categories are not mutually exclusive, and there may be some overlap in the application areas for each type of device.

**Table 2.** Types of IoMT devices and their application areas with examples.

| Type of Device | Application Area | Examples |
|---|---|---|
| Wearable Devices | Remote Patient Monitoring, Chronic Disease Management, Fitness Tracking | Smartwatches, Fitness Trackers, Continuous Glucose Monitors (CGMs) |
| Implantable Devices | Chronic Disease Management, Patient Monitoring | Pacemakers, Neuro-stimulators, Implantable Cardioverter Defibrillators (ICDs) |
| Ingestible Devices | Patient Monitoring, Medication Adherence | Smart Pills, Digestible Sensors |
| Smart Medical Equipment | Hospital Workflow Optimization, Remote Monitoring | Smart Beds, Smart Infusion Pumps, Remote Vital Sign Monitors |
| Health and Wellness Devices | Health and Wellness Tracking, Disease Prevention | Smart Scales, Blood Pressure Monitors, Digital Thermometers |

*5.2. Types of IoMT*

Each type of IoMT device and application has its own set of benefits and challenges. However, when used together in a cohesive system, they provide a powerful tool for improving patient outcomes and delivering personalized healthcare. Implantable medical devices (IMDs) and Internet of Wearable Devices (IoWDs) are two types of connected medical devices that fall under the umbrella of IoMT [42].

- Implantable medical devices

IMDs are medical devices that are implanted inside the body, such as pacemakers, defibrillators, and neuro-stimulators. These devices provide continuous monitoring and treatment for chronic conditions. They collect data on a patient's vital signs, heart rate, and other health parameters, which are transmitted to healthcare professionals for analysis. They can also be programmed to deliver therapeutic interventions, such as electrical impulses to treat chronic pain or irregular heart rhythms. One of the benefits of IMDs is that they improve patient outcomes by providing continuous monitoring and treatment for chronic conditions. They also reduce the need for hospital visits and enable patients to manage their conditions more effectively from home. However, there are also concerns about the security and privacy of patient data collected by IMDs, as well as the risks associated with surgical implantation.

- Internet of wearable devices

IoWDs, on the other hand, are wearable medical devices that are worn on the body, such as smartwatches, fitness trackers, and health monitors. These devices track a variety of health parameters, including heart rate, blood pressure, and sleep patterns. They transmit data to healthcare professionals for analysis and monitoring, enabling them to make informed decisions about patient care. IoWDs have the advantage of being non-invasive and easy to use. They also provide patients with real-time feedback on their health status and encourage them to make healthier lifestyle choices. However, there are concerns about the accuracy and reliability of IoWDs, as well as the potential for data breaches and cyber-attacks.

Overall, both IMDs and IoWDs have the potential to revolutionize healthcare delivery by providing real-time monitoring and personalized treatment plans. As the IoMT ecosystem continues to evolve, we can expect to see more innovative uses of these devices and applications in healthcare. The main difference between IMDs and IoWDs is that IMDs are implanted inside the body, while IoWDs are worn on the body. IMDs are typically used for more serious medical conditions that require continuous monitoring and treatment, while IoWDs are used for more general health and wellness monitoring. However, concerns exist regarding the security and privacy of patient data collected by these devices, as well as the accuracy and reliability of the data.

### 5.3. IoMT Systems Architecture

The IoMT architecture serves as the foundation for the seamless integration of medical devices, sensors, and healthcare systems with IoT technology. The IoMT architecture is composed of various layers, each with its own functions and components. In this section, we will discuss the different layers of the IoMT architecture as depicted in Figure 7.

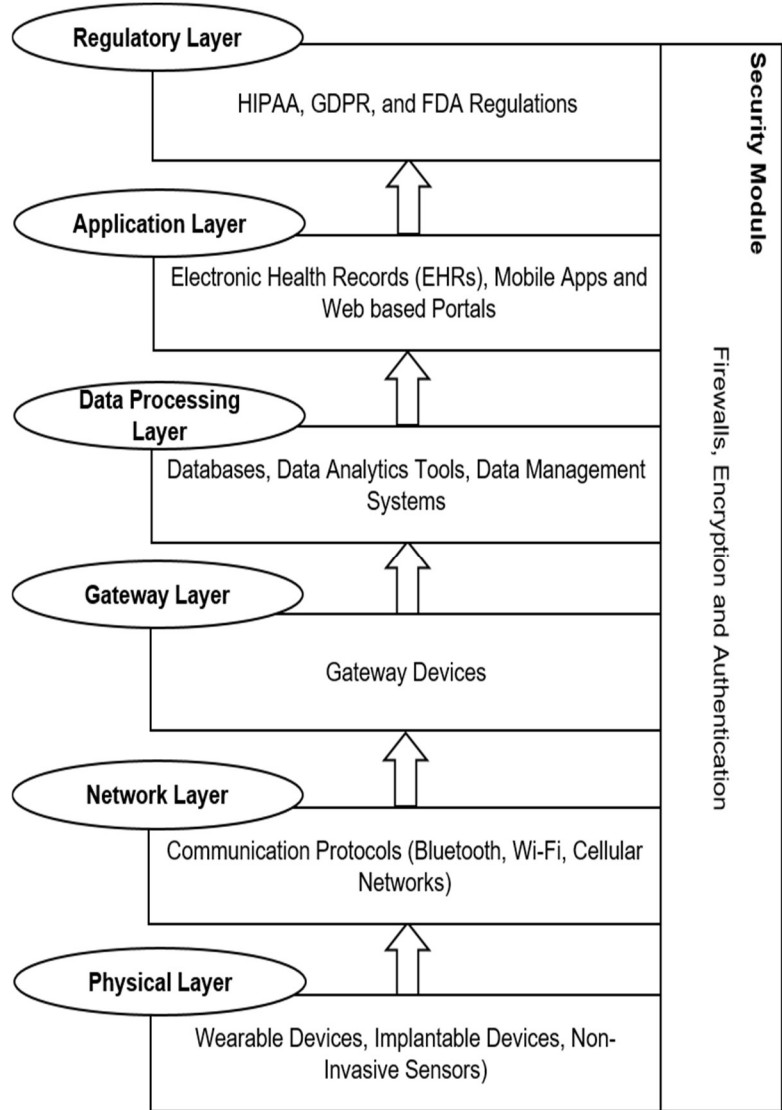

**Figure 7.** Internet of Medical Things System Architecture.

### 5.3.1. Physical Layer

The physical layer serves as the foundation of the IoMT architecture, encompassing a range of devices and sensors utilized for data collection from patients. It consists of wearable, implantable, or non-invasive devices that collect data from patients. These devices can include wearables such as smartwatches, implantable devices such as pacemakers, or non-invasive sensors for monitoring vital signs. The physical layer is responsible for gathering and transmitting the patient's health data to the rest of the IoMT architecture.

### 5.3.2. Network Layer

The network layer handles communication between devices and systems in the IoMT architecture. It utilizes various communication protocols such as Bluetooth, Wi-Fi, and cellular networks to ensure secure and efficient data transmission. The network layer

enables connectivity and data exchange between different devices and systems within the IoMT ecosystem.

### 5.3.3. Gateway Layer

The gateway layer in the IoMT architecture plays a crucial role in facilitating communication and data transmission between the physical layer, where devices and sensors are located, and the data layer, where data is processed and stored. It acts as a bridge connecting the two layers and ensures the secure and efficient exchange of information.

### 5.3.4. Data Processing Layer

The data layer assumes responsibility for storing and processing the data acquired by the physical layer. It comprises various components such as databases, data analytics tools, and data management systems. The data layer ensures the security, accuracy, and easy accessibility of the data for healthcare professionals.

### 5.3.5. Security Layer

The security layer plays a crucial role in safeguarding the IoMT system against cyber threats and ensuring its overall security. It encompasses a range of components, including firewalls, encryption mechanisms, and authentication systems. The security layer's primary objective is to ensure the protection and confidentiality of the data collected by the IoMT system, mitigating the risk of unauthorized access or theft.

### 5.3.6. Application Layer

The application layer plays a vital role in granting healthcare professionals access to the data collected by the IoMT system. It encompasses a range of applications, including electronic health records (EHRs), mobile apps, and web-based portals. Through the application layer, healthcare professionals can monitor patients' health status, offer real-time feedback, and make informed decisions regarding diagnosis and treatment.

### 5.3.7. Regulatory Layer

The regulatory layer plays a crucial role in ensuring compliance with relevant regulations and standards within the IoMT system. It encompasses various components, including auditing systems, compliance frameworks, and certification programs. The primary objective of the regulatory layer is to ensure that the IoMT system adheres to regulations such as HIPAA, GDPR, and FDA regulations, thereby ensuring the system's safety, effectiveness, and compliance with healthcare requirements.

In conclusion, the IoMT architecture represents a sophisticated system that facilitates the integration of medical devices, sensors, and healthcare systems with IoT technology. When designing the IoMT architecture, several crucial considerations must be taken into account, including security, privacy, scalability, and interoperability. Security measures need to be implemented to safeguard patient data against cyber threats, while privacy regulations should be adhered to in order to maintain the confidentiality of patient information. Scalability is essential to accommodate the increasing number of connected medical devices and sensors within the network, and interoperability is critical to enabling seamless exchange and analysis of data collected from diverse devices.

## 6. Enabling Technologies in Healthcare

Technology-enabled healthcare is a way to transform healthcare delivery in cities, providing improved access to healthcare services, better health outcomes, and improved sustainability. This is achieved by using emerging technologies, such as IoT, cloud computing, fog computing, edge computing, big data, and blockchain, as depicted in Figure 8. These technologies help improve healthcare services and information by allowing real-time data sharing and communication between healthcare providers, patients, and other stakeholders. Some of the enabling technologies are discussed below:

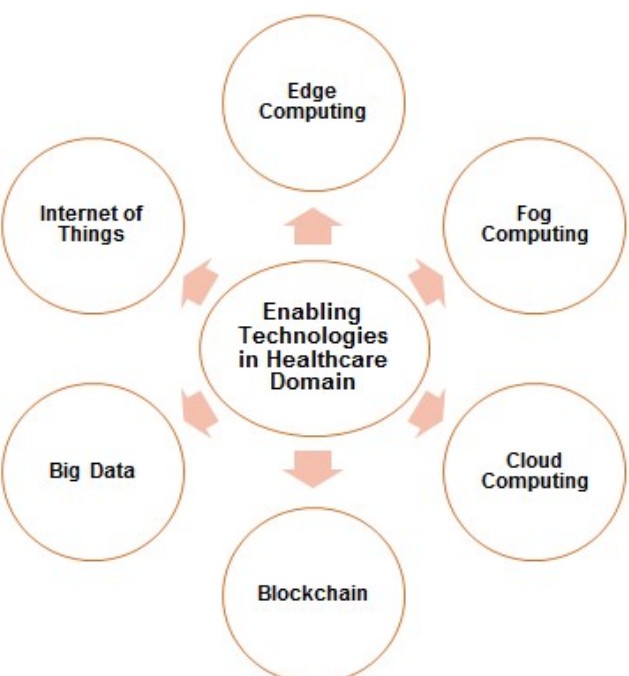

**Figure 8.** Enabling Technologies in Healthcare Domain.

### 6.1. Internet of Things

The IoT is a technology that allows medical devices to collect, transmit, and analyze data in real-time [55]. This helps healthcare providers make real-time, data-driven decisions and improve the quality of patient care. It is used for remote patient monitoring, allowing patients to receive care and monitoring without the need to visit the doctor's office. Using IoT, medical devices and specialized care services are identified, detected, and authenticated. Globally, the term 'IoT' refers to a network of physical devices that are connected to the internet in order to collect and share data. With the help of wireless sensors, the IoT network gathers and shares data. IoT enables the delivery of personalized, affordable, and cost-effective services in digital healthcare. Using a wireless system, IoT streamlines data collection and communication, helping doctors and nurses save time and effort in accessing and analyzing patient information. This enables medical practitioners to have real-time access to comprehensive patient information, empowering them to make timely and well-informed decisions. In addition, remote patient monitoring reduces the need for frequent in-person visits and manual data collection, resulting in cost savings for the healthcare system.

### 6.2. Healthcare Cloud, Fog, and Edge Computing

Healthcare cloud computing utilizes cloud computing technologies in the healthcare industry. Cloud computing is a model that enables ubiquitous, convenient, on-demand access to a shared pool of configurable resources, such as computer networks, servers, storage, applications, and services. These resources are rapidly provisioned and released with minimal management effort or service provider interaction. It is a technology that facilitates the storage, processing, and sharing of medical data in the cloud [56,57]. Cloud computing in healthcare involves the utilization of remote computing resources for data storage and processing, offering healthcare organizations a secure and cost-effective method to store and manage large volumes of data. Additionally, it provides them with access to data from anywhere at any time, facilitating easier collaboration on projects. The adoption of cloud computing also helps reduce IT costs by eliminating the need for additional hardware and software, while allowing scalability as required. Furthermore, it fosters seamless collaboration between healthcare organizations, leading to improved patient care.

By adopting cloud computing, healthcare organizations can securely store and access vast amounts of data in an efficient and cost-effective manner.

Fog computing is a decentralized computing architecture that incorporates cloud computing into the IoT. It enables data processing close to the source of data collection rather than in a centralized cloud. This approach allows healthcare organizations to process data closer to where it is created, resulting in improved response times and reduced latency. Additionally, it helps alleviate the strain on data centers, leading to cost reductions. Moreover, fog computing enables quick and secure data sharing with other organizations, ultimately enhancing patient care. Furthermore, fog computing provides healthcare organizations with the flexibility to scale their data processing operations as needed, facilitating rapid adaptation to changing demands [58–61].

Edge computing minimizes network latency by reducing the amount of data sent over the network. Data processing occurs locally instead of being transmitted to a cloud-based server, resulting in faster processing times. This approach enables healthcare providers to reduce costs associated with data processing and enhance the security of patient data. Additionally, it allows for the storage of large data volumes closer to the network edge, enabling faster and more secure data processing [62,63]. Healthcare cloud computing, fog computing, and edge computing assist healthcare providers in processing large volumes of data quickly, securely, and cost-effectively. These technologies play a significant role in Healthcare 4.0, positively impacting healthcare research and service improvement. They enhance quality, increase affordability for a larger population, and improve patient outcomes.

### 6.3. Big Data

Big data refers to data sets that are extremely large and complex, making it challenging to process them using traditional data-processing applications [64]. In healthcare, big data encompasses the collection, storage, and analysis of vast amounts of data to gain insights into patient care and health outcomes. Its potential to revolutionize the healthcare industry lies in its ability to enhance patient care and improve operational efficiency. By leveraging data-driven analytics and machine learning, healthcare organizations can obtain unprecedented insights into patient care, enabling informed decisions regarding treatment plans, diagnoses, and medication choices. Furthermore, big data in healthcare facilitates the identification of trends and patterns in patient care, leading to improved outcomes and cost reduction. It empowers healthcare professionals to make more informed decisions, thereby enhancing the quality of care provided. However, there are concerns raised by experts regarding the use of big data in healthcare. One concern is the potential for unfair targeting of certain groups, such as individuals with pre-existing conditions. Another worry is the possibility of manipulating patients into making decisions that are not in their best interests. The increasing availability of information through various online platforms, including social networking sites, feeds, and online discussion forums, is dramatically changing medicine [65]. Real-time monitoring of individuals' health status generates vast amounts of data from personal devices, wireless sensors, and mobile communication technologies.

### 6.4. Blockchain Technology

Blockchain technology possesses the potential to revolutionize healthcare by enabling the secure storage and transfer of patient data. This technology is utilized to establish a more secure, accurate, and transparent system for maintaining and sharing medical records and other health-related data. It contributes to the reduction of medical errors, enhances data security, and facilitates the tracking and analysis of patient health trends. By employing blockchain technology, healthcare data can be securely stored and remain immutable. It enhances the precision of medical records and diminishes medical errors. Healthcare data, including patient records, medical bills, clinical trial results, and other pertinent information, can be stored using blockchain technology. Furthermore, it holds the potential to streamline administrative tasks, reduce costs, and improve the quality of care. Additionally, blockchain technology safeguards patient privacy, ensuring that only authorized users can access

sensitive health information. A significant advantage of utilizing blockchain technology in healthcare is the elimination of intermediaries such as insurance companies, for mediating transactions between patients and healthcare providers. This reduction in intermediaries decreases costs and enhances the security and transparency of transactions. Moreover, it facilitates more efficient data sharing between healthcare providers, leading to more accurate diagnoses and effective treatments. Data provenance, robustness, decentralized management, security, and privacy are considered vital aspects of blockchain technology in health and medical services [66]. Although blockchain technology has the potential to revolutionize healthcare, its adoption also entails risks and challenges. The technology is still in its early stages of development, and its scalability remains uncertain. Additionally, implementing blockchain technology in healthcare requires significant investment from healthcare organizations, with no guarantee of success.

## 7. IoMT Healthcare Services and Applications

Smart healthcare services use modern technology such as AI, machine learning, and IoT to automate tasks, collect patient data, and improve the accuracy and effectiveness of patient care. This helps to reduce medical errors and improve patient outcomes. This makes healthcare services more efficient, cost-effective, and reliable, providing patients with better care and quality of life. There are many use cases for the IoMT for patients, medical professionals, researchers, and insurers. IoMT also enables remote patient monitoring, which can reduce healthcare costs and improve outcomes. These include immediate medical help, analysis of data, prescription management, augmented operations, monitoring patients, employees, and inventory, and many more, as illustrated in Figure 9. Some of the services [67] and applications are discussed below:

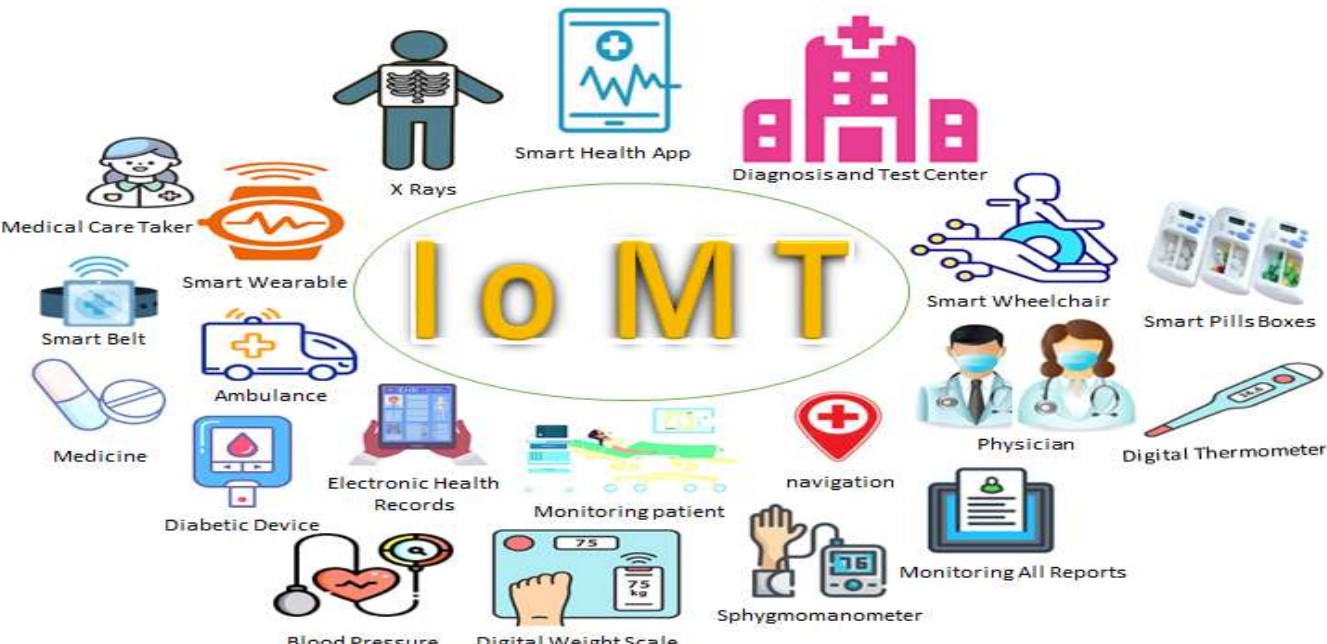

**Figure 9.** Services and applications of internet of medical things (IoMT).

### 7.1. Drones

Drones provide a cost-effective way to deliver medical supplies to remote locations and facilitate the transportation of samples and equipment for medical diagnoses. They also enable remote medical consultations, monitor health conditions, and offer swift response times in emergency situations. Internet-based drones have a wide range of applications, including surveillance, mass testing, public announcements, medical diagnoses, medical deliveries, and disinfection [68]. These drones are utilized for delivering medical supplies to remote regions, monitoring and assisting in hazardous areas, responding rapidly to

emergencies, and providing essential medical care. They also play a vital role in surveillance and mass testing efforts, facilitate medical announcements, assist in medical diagnoses, and disinfect areas to prevent the spread of diseases and infections. Diagnostic drones, such as thermal-imaging drones, enable remote monitoring and detection of infected cases by observing their temperature, heart rate, and symptoms such as sneezing or coughing [69,70].

### 7.2. Robotics Medical Applications

Robotics-based smart healthcare applications significantly improve the accuracy of diagnosis and treatment. They enhance patient monitoring and enable faster response times to medical emergencies. Additionally, they automate tedious and repetitive tasks, allowing healthcare workers to focus on more complex issues. A humanoid robot reduces physical interactions between healthcare workers and infected individuals [71]. Among the many tasks robots assist with are food preparation, medication administration, and cleaning and disinfection of medical facilities [72]. This reduces the time and labor required for these tasks, improving efficiency and reducing the risk of human error. Although robotics-based smart healthcare applications offer great potential, there are also some risks. Moreover, robots perform tasks deemed too dangerous for humans, such as cleaning hazardous areas [73]. Another risk is excessive reliance on technology at the expense of human interaction, potentially compromising the quality of patient care.

### 7.3. Assisted Living (in Home Care)

Assisted living is a form of long-term care for elderly and disabled individuals who need help with daily activities such as bathing, dressing, and medication. It allows them to remain in their own home while still receiving the care they need. This approach avoids unnecessary hospitalization by enabling patients to stay in a familiar environment [74]. These telepresence and videoconferencing robotic solutions help ensure patient safety and monitoring while also enabling them to maintain contact with family and healthcare professionals [75]. Additionally, these solutions serve to provide educational or therapeutic activities for older individuals without requiring them to learn complex technological systems [76]. Ambient assisted living solutions are designed to be simpler and more intuitive, making them easier to use and understand for those who may not be as familiar with the technology. This enables individuals who may struggle with more complex systems to still access the benefits of technology [77]. The use of AI methodologies for ambient intelligence systems in the healthcare domain helps process the vast amount of monitoring data efficiently and scale effectively with the number of ambient-assisted patients. These technologies will enable healthcare centers to receive automatic alerts for observation and emergency assistance. Cloud and Fog technologies can support healthcare and provide on-demand infrastructure [78,79].

### 7.4. Community and Children's Healthcare Services

These services provide a range of healthcare services for children in the local community, including preventative care, medical check-ups, vaccinations, and mental health support. They also offer education on nutrition and health topics, as well as access to resources for families in need. In healthcare services, a cooperative IoT platform has been proposed and found to be energy-efficient [80]. The concept of a community medical network is presented in [81]. This will help reduce the stigma associated with mental health issues and provide more support for parents and children dealing with them. It will also ensure that children have access to the resources they need to address any underlying mental health issues they may have. An interactive totem is proposed for placement in a pediatric ward to provide services for hospitalized children aimed at educating, entertaining, and empowering them [82]. IoT-based m-health services are presented to encourage children to adopt healthy nutritional habits with their teachers and parents [83].

### 7.5. Personalized Healthcare

Personalized healthcare involves the use of technology to individualize medical treatments for each patient [84,85]. This is achieved by collecting data on a patient's medical history, lifestyle, and genetics and then using that data to create a customized treatment plan that is tailored to the specific needs of each patient. By utilizing this approach, healthcare professionals gain the ability to more accurately diagnose illnesses, identify potential risks, and design treatments that are tailored to the individual's unique needs. By implementing this personalized approach, healthcare professionals provide their patients with the most effective and individualized care possible, resulting in enhanced patient outcomes and satisfaction [86]. Thus, big data analytics play a crucial role in the implementation of individualized healthcare, both for individuals and populations [87,88]. On the other hand, there are also some drawbacks to personalized healthcare. First, it can be expensive to implement, and second, it may lead to healthcare providers becoming too reliant on technology, which could ultimately compromise patient care.

### 7.6. Rehabilitation

Smart healthcare services provide patients with access to personalized rehabilitation plans that are tailored to their individual needs. This helps optimize the effectiveness of their rehabilitation and reduces the required time for completion. A home-based rehabilitation program is expected to decrease healthcare costs and enhance patient quality of life. IoT-based smart rehabilitation systems can be developed using an ontology-based automated design method [89]. IoT-based technologies can support effective remote consultations in comprehensive rehabilitation. As a field of medicine dedicated to enhancing and restoring functional ability and quality of life for individuals with disabilities or impairments, physical medicine holds significant importance. Numerous IoT-based rehabilitation systems have been developed, including an integrated application system for correctional facilities, a medical rehabilitation system for smart cities [90,91], and a language-training system for children with autism [92].

### 7.7. Chronic Disease Management, Medication Management, Telemedicine, and Drug Delivery

IoMT helps manage chronic diseases by providing continuous monitoring of patients' health status and enabling early intervention. A smartphone-based platform that utilizes sensors to monitor patients with chronic obstructive pulmonary disease (COPD) was able to detect early signs of exacerbation and alert healthcare professionals, resulting in reduced hospitalizations and improved patient outcomes. Medication management systems ensure the accurate and safe delivery of medications to patients. They also provide a way to document and store patient records, ensuring that the right medications are prescribed and administered. Additionally, they ensure that patients receive the correct medications at the correct dosages. Medicine boxes are intelligently packaged for IoT-based use in medication management [93]. RFID tags are used in an e-health service architecture for medication control through an IoT network [94]. These systems also reduce the time healthcare professionals spend filling prescriptions and streamline the process of ordering and administering medications. By automating certain processes, they reduce costs and improve efficiency.

Telemedicine also significantly improves the quality of medical care, as medical practitioners have access to medical data from various locations and can deliver better diagnoses and treatment plans for their patients. This, in turn, leads to a reduction in medical costs as doctors provide more efficient and accurate medical advice. IoMT improves drug delivery by providing real-time monitoring of drug levels and enabling personalized dosing. A smart pill that is ingested and controlled wirelessly allows healthcare professionals to monitor drug levels and adjust dosages as needed. This technology has the potential to improve medication adherence and reduce adverse effects.

### 7.8. Wheelchair Management

Wheelchair management helps reduce the risk of falls and injuries and improves the patient experience by providing a safe and efficient way to move patients around. It also helps reduce the time required to move patients, freeing up medical staff for other tasks. Proper safety protocols must be in place, such as ensuring wheelchair brakes are engaged and patients are securely fastened in the chair. Trained staff are necessary to operate and maneuver wheelchairs correctly and identify potential safety issues. In [95], an IoT-based wheelchair healthcare system is proposed, while ref. [96] implements a peer-to-peer (P2P) and IoT-based medical support system. By monitoring the vital signs of the person seated in the chair and collecting information about their surroundings, these devices enable users to evaluate the accessibility of a location.

### 7.9. Smartphones Services

Smartphone services enable healthcare systems to provide better patient care, remotely monitor health status, and increase patient engagement. Mobile apps allow patients to easily access health records, schedule appointments, and communicate with doctors. Smartphone-controlled sensor services support patient monitoring, telemedicine, and remote doctor visits. They provide quick and secure access to medical records, promoting seamless information sharing among healthcare providers and enhancing patient care while reducing costs. Versatile healthcare functionalities are achieved on smartphones with compatible hardware and software products. Diagnostic apps offer access to information on diagnostics and treatments, while medical education apps provide tutorials, training materials, surgical demonstrations, and medical illustrations. In ref. [97] presents image analysis algorithms for non-contact measurements in healthcare applications. However, implementing IoT smartphones in healthcare systems presents challenges that need to be addressed. Data security and privacy are major concerns in light of increasing data breaches and cyber-attacks, requiring robust security measures to protect patient information. Standardization among IoT devices is another challenge, hindering interoperability and data exchange between different devices and systems. Additionally, not all individuals are interested in fitness and health tracking, and some may find constant monitoring invasive or uncomfortable. Moreover, there are alternative devices such as Fitbit's Zip or One that cater to fitness tracking without relying on a smartphone.

### 7.10. Remote Patient Monitoring

IoMT devices are primarily utilized for remote patient monitoring in healthcare. This technology enables real-time monitoring of patients, even when they are outside of a healthcare facility. Consequently, healthcare professionals can assess a patient's health quickly and accurately, providing timely and appropriate care [98]. Health metrics such as heart rate, blood pressure, temperature, and more are automatically collected by IoT devices from patients who are not physically present in healthcare facilities. As a result, patients are relieved from the need to travel to their providers or manually collect this information. The collected data is then transmitted to a software application that can be viewed by healthcare professionals and/or patients. Algorithms are utilized to analyze the data, generate alerts, and recommend treatments. While remote patient monitoring holds the potential to improve patient outcomes, it is crucial to address privacy and security concerns. Additionally, there is concern that insurance companies could misuse patient data to discriminate against individuals with pre-existing conditions. One major challenge is the risk of hackers gaining unauthorized access to patient data transmitted wirelessly.

### 7.11. Glucose Monitoring in Healthcare

Monitoring glucose levels is important for people with diabetes to help them manage their blood sugar levels. It can also be used to monitor other conditions, such as hypoglycemia, which can be caused by certain medications or illnesses. By monitoring glucose levels closely, individuals with diabetes are proactive in managing their health and

avoiding the risk of developing potentially serious complications. It takes a great deal of time to measure and record glucose levels manually, and this is not only inconvenient; it also only reports a patient's glucose level at the moment of the test. In cases where levels fluctuate widely, periodic testing may not be sufficient to detect problems. By providing continuous, automatic glucose monitoring to patients, IoT devices help address these challenges [99]. Patients are alerted when their glucose levels are abnormal when glucose monitoring devices replace manual record-keeping. For instance, if a person notices their glucose levels starting to decline, they take steps to increase them before they become dangerously low. However, there are potential risks associated with continuous glucose monitoring. Improper transmission or misinterpretation of data from the device can result in false alarms or inaccurate readings. This may cause unnecessary anxiety for patients or, worse, lead to dangerous situations if patients do not take appropriate actions in response to the alarms.

### 7.12. Heart Rate Monitoring

Heart rate monitoring helps measure the intensity of physical activity and determine the effectiveness of a workout. It provides insight into a person's overall health, as changes in heart rate indicate potential health issues. By tracking a person's heart rate during physical activity, it determines the effort exerted and the required rest between sets. Moreover, changes in heart rate can indicate if someone is overexerting themselves, which may lead to injury or illness. Monitoring heart rate also helps detect underlying health issues [100]. Overall, tracking heart rate during exercise offers valuable information for ensuring a safe and beneficial workout routine. However, some individuals may feel uncomfortable with their heart rate being monitored during exercise. Additionally, privacy concerns may arise if heart rate data is collected and stored.

### 7.13. Hand Hygiene Monitoring

This involves tracking the frequency and effectiveness of handwashing with soap and water by healthcare workers and patients. It plays a crucial role in reducing the spread of infection and understanding the efficacy of hand hygiene practices. It is important because inadequate handwashing by healthcare workers or patients can facilitate the transmission of germs and bacteria, leading to the spread of illnesses and infections. Therefore, monitoring and tracking the hand hygiene practices of healthcare workers and patients is crucial. It is essential to provide education and training on proper handwashing techniques and ensure consistent and correct adherence to hand hygiene practices [101]. Many hospitals today use IoT devices to remind patients to sanitize their hands upon entering hospital rooms. These devices also provide recommendations on hand sanitization techniques based on specific risks for individual patients. One limitation of these devices is that they cannot physically wash the hands of the user. Nonetheless, research suggests that these devices significantly reduce hospital infection rates by more than 60 percent.

### 7.14. Tracking of Depression and Moods

Tracking depression and moods helps individuals identify potential triggers and inform treatment options to better manage symptoms and improve their overall well-being [102]. This process enables individuals to better understand their own mental health and recognize patterns in their behavior that may indicate a need for medical intervention. Additionally, tracking depression and moods helps clinicians identify any changes over time and adjust treatments accordingly. By using an IoT device, such as a fitness tracker or smartphone, to track individuals' moods, we can identify patterns in their behavior that point to depression or other mental health issues. With the data collected, users work with a mental health professional to develop strategies to manage their moods or seek help if necessary. This provides a better understanding of how depression affects individuals over time and can be utilized to develop better treatments or interventions for those suffering from the condition. These data create personalized care plans, alert healthcare providers to sudden changes in

mental state, and provide patients with real-time feedback and guidance on how to cope with their emotions.

### 7.15. Monitoring Parkinson's Disease

By monitoring Parkinson's disease progression, doctors gain a better understanding of how the disease is affecting an individual and are able to adjust treatment accordingly. This ensures that the patient receives the best possible care and effectively manages their symptoms. Monitoring disease progression also helps doctors comprehend how the disease is changing over time, providing valuable insight into the optimal treatment approaches [103]. This enables them to identify new therapies and treatments that may be more effective at managing the disease than the current ones being used. An IoT device is utilized to monitor a person's movements and any changes in their behavior that may indicate the onset of Parkinson's disease. It is also used to monitor changes in medication and other lifestyle habits that can influence the disease. By tracking these changes, doctors and patients are able to adjust therapies and treatments as needed to ensure the most favorable outcome. This active monitoring and adjustment of treatment plans helps optimize the management of the disease and achieve the best possible results.

### 7.16. Connected Inhalers/Ingestible Sensors/Connected Contact Lenses

In conditions such as asthma, attacks often occur suddenly and without warning. Using an IoT-connected inhaler, healthcare providers are able to track the frequency of attacks and collect data from the environment to identify what triggers them. This data is used to create personalized treatment plans that help patients identify and avoid triggers as well as better manage their condition. The inhalers also have the capability to alert healthcare providers when an attack is imminent, allowing them to intervene if necessary. However, some argue that IoT-connected inhalers pose privacy concerns. The constant monitoring of patients' inhaler usage could potentially lead to over-diagnosis and unnecessary treatments. Additionally, the data collected by the inhalers could be misused to discriminate against patients with asthma or other respiratory conditions.

Furthermore, ingestible sensors are small devices that are swallowed and track the body's internal processes. They measure a range of biometric data, such as temperature, pH levels, and metabolic rate, and transmit this data wirelessly to a connected device. These sensors have the potential to revolutionize healthcare by facilitating real-time monitoring of a patient's health. They detect changes in body temperature, heart rate, and other vital signs that aid in the quick and accurate diagnosis of medical problems [104]. For instance, a temperature sensor embedded in a patch worn on the skin detects a fever in a patient, alerting doctors and caretakers to take action. However, there are also potential dangers associated with these sensors. If the data collected by the sensors is not properly secured, unauthorized individuals could access it. This could lead to the unauthorized disclosure of sensitive information about a patient's health. However, if the sensors are not used properly, they could provide false readings that may result in misdiagnoses.

Connected contact lenses are contact lenses that are equipped with sensors and other technology that allows them to connect to other devices, such as smartphones. These lenses are embedded with tiny sensors that detect physiological changes in the wearer's body and transmit that data to a connected device. This data is then used to monitor and manage various health conditions and provide diagnoses that are more accurate. By collecting this data and using it to inform diagnoses, it provides healthcare providers with more accurate and timely information, which leads to treatments that are more effective.

## 8. Open Research Challenges and Research Directions

In recent years, many researchers have been working on designing and implementing healthcare services and addressing related technical and architectural issues. Along with the research concerns in the literature, there are several challenges and open issues that need to be addressed. This paper discusses some of the major challenges, as depicted

in Figure 10. These challenges include developing new approaches to integrate diverse data sources, understanding user behavior and usage patterns, developing personalized healthcare services, and advancing technologies for monitoring and diagnosis [105–107]. Furthermore, research directions are focused on predictive analytics, machine learning approaches for diagnosis, and visualization techniques for presenting healthcare data.

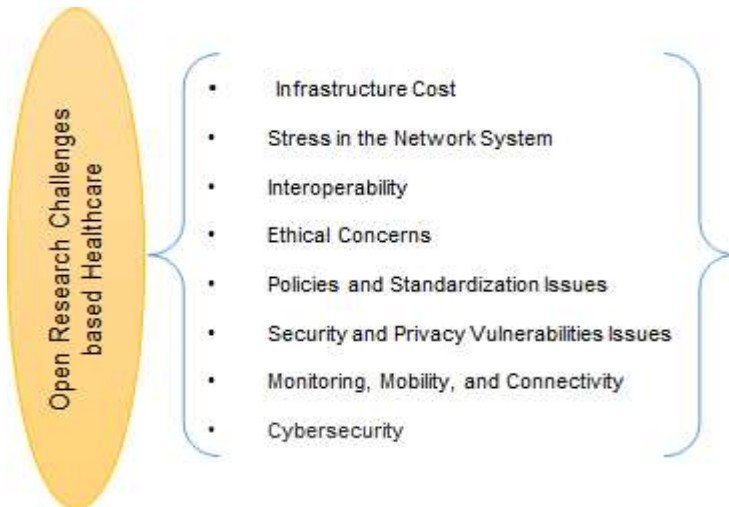

**Figure 10.** Research Challenges in Healthcare Domain.

### 8.1. Cost of Infrastructure

The implementation of IoMT is expensive due to the high cost of acquiring the necessary hardware and software equipment, building an application, and maintaining and storing the generated data. These initial costs pose a challenge for the implementation of IoMT in the early stages. However, it is important to consider the long-term benefits of IoMT, such as reduced overall medical costs, improved patient outcomes, and increased efficiency in healthcare delivery. Despite the initial cost, the investment in IoMT ultimately leads to significant cost savings and improved patient care, making it a worthwhile investment for healthcare organizations [108].

### 8.2. Stress in the Network System

Healthcare organizations currently rely on Wi-Fi and other similar technologies to connect IoMT devices within hospitals. However, many organizations lack the necessary network infrastructure to effectively integrate and utilize these devices. To fully leverage IoMT technology, healthcare organizations must first invest in the development of this infrastructure. In simpler terms, hospitals currently use Wi-Fi to connect IoMT devices, but many need to upgrade their network infrastructure to fully utilize IoMT technology.

### 8.3. Interoperability

As the IoMT ecosystem continues to grow, there is a need for standardization and interoperability among different devices and platforms. This will allow healthcare professionals to access patient data from a variety of sources, enabling them to make informed decisions about patient care.

### 8.4. Ethical Concerns

IoMT also raises ethical concerns about the use of patient data. As medical devices collect more data on patients, there is a risk of misuse or unauthorized use of the data for purposes other than healthcare. This raises questions about patient consent and the ownership of medical data.

### 8.5. Policies and Standardization Issues

Policies and standardization help improve the quality of healthcare services by providing a framework for the consistent delivery of care. They also help to reduce costs and errors and increase patient safety by ensuring that all healthcare providers are following the same protocols. It ensures that healthcare providers deliver quality care in a consistent and cost-effective manner. Standardization helps to improve patient safety, reduce medical errors, streamline processes, and enhance overall quality. It has been estimated that approximately 51% of medical devices currently in production are governed by Food and Drug Administration (FDA) rules, which disrupt medical delivery because they affect IoMT efficiency [109].

Various value-added services, such as electronic health records and more, also require standardization. To achieve this, organizations such as the Information Technology and Innovation Foundation (IETF), the Internet Protocol for Smart Objects (IPSO) Alliance, and the European Telecommunication Standards Institute (ETSI) collaborate through IoT technology working groups. These groups work alongside m-health and e-health organizations and IoT researchers to standardize IoT-based healthcare services. Additionally, the healthcare sector is fragmented, with different stakeholders having diverse interests and objectives, which makes implementing a unified policy challenging. Despite these challenges, policies and standardization in healthcare offer numerous benefits. They help ensure that patients receive consistent, high-quality care regardless of the treatment location. Furthermore, these policies enhance communication and coordination among healthcare providers, leading to improved patient outcomes.

### 8.6. Security and Privacy Vulnerabilities Issues

Security and privacy are crucial in healthcare services to ensure the safety and confidentiality of patient data as well as prevent the unauthorized sharing of sensitive information. Healthcare organizations must ensure that patient data remains secure and private, as mandated by HIPAA regulations [110]. HIPAA regulations require that healthcare service providers implement a variety of measures to protect patient data, such as encryption and access control systems. Healthcare providers must also have a written policy in place to ensure that patient data is protected and that any breaches of the policy are reported immediately. To ensure the confidentiality of medical information, unauthorized users are prevented from accessing it. An IoT health device uses authentication to verify the identity of its peers and ensure secure communication. A network service or resource can only be accessed by authorized nodes if they are authorized. Even if a fault occurs, a security scheme should continue to provide its respective security services.

These challenges have a significant impact on the quality of patient care and need to be addressed through effective policies and procedures. For instance, it is crucial to store patient data in secure databases that are regularly updated with the latest security patches and monitored for any suspicious activities. However, some argue that these challenges may not be as significant as they appear and that the benefits of digital health outweigh the risks. For example, digital health improves patient outcomes by providing real-time data that is used to make informed decisions about treatment. It also enhances transparency and communication between patients and healthcare providers.

### 8.7. Monitoring, Mobility, and Connectivity

Monitoring, mobility, and connectivity are essential elements of healthcare, as they enable services to be accessible to more people and facilitate tracking and monitoring of patient health. This allows healthcare providers to monitor their patients from anywhere and provide better care. Connectivity also enables patients to remotely access their medical records and data, making it more convenient to seek medical advice and treatments. Additionally, mobility facilitates healthcare providers to reach more people, as they are not limited to being in the same location as their patients. In a connected smart health system, both patients and medical staff should have the ability to move freely between

multiple locations. However, the presence of several wireless networks operating in close proximity leads to collisions, particularly when mobility is involved. Healthcare delivery faces disruptions and reduced performance, which may lead to disastrous situations. Therefore, it is essential to ensure the accurate operation of medical devices by connecting them through various wireless technologies. The challenges associated with monitoring, connectivity, and mobility in IoT/mobile applications are numerous. Despite significant progress in sensors and IoT technology, achieving the same level of accuracy as hospital-grade devices while maintaining energy efficiency and wearability remains a challenge. To address this, we need new technologies that are cost-aware, powerful, fully connected, and efficient. The evolution and innovation of smart and connected healthcare are expected to provide efficient solutions and services to patients, paving the way for next-generation healthcare. Research and development efforts are necessary to ensure data security and user privacy. Additionally, there is a need for research to improve methods for monitoring healthcare services and enhancing their mobility and connectivity.

### *8.8. Cybersecurity*

Cybersecurity is one of the major challenges facing IoMT. As IoMT devices become more interconnected and accessible, they become more vulnerable to cyber-attacks. It refers to the interconnected network of medical devices, sensors, and other healthcare technology that is connected to the internet. While IoMT has the potential to revolutionize healthcare and improve patient outcomes, it also presents significant cybersecurity challenges. Medical devices and healthcare systems are often vulnerable to cyber-attacks, which can result in the theft of sensitive patient data, the manipulation of medical records, or even physical harm to patients. This is because many medical devices and systems were not designed with cybersecurity in mind and are often not regularly updated with security patches. Additionally, many healthcare providers have not yet fully integrated cybersecurity into their overall risk management strategies, which leaves them vulnerable to cyber threats. As a result, it is important for healthcare providers and medical device manufacturers to prioritize cybersecurity in order to ensure the safety and security of patient data and the integrity of medical devices and systems. This may involve implementing strong encryption and authentication protocols, regularly updating and patching systems, and investing in cybersecurity training and awareness for staff.

### 9. Conclusions

Across the globe, researchers, scientists, industries, and governments are exploring the healthcare sector, which has captured their attention. The smart and connected healthcare system in smart cities involves leveraging technology to enhance the quality of healthcare delivery, from diagnosis to treatment. It encompasses the utilization of emerging technologies, such as artificial intelligence, virtual reality, and other advanced technologies, to streamline the healthcare system and enhance its efficiency. Moreover, the IoMT holds the potential to revolutionize healthcare delivery by enabling real-time patient monitoring, improving diagnosis, and enabling personalized treatment plans. However, the widespread adoption of IoMT technologies also presents challenges concerning data privacy and security, interoperability, and regulatory compliance. Tackling these challenges is crucial to fully realizing the potential of the IoMT and improving healthcare outcomes for patients worldwide. This paper explores intelligent, personalized healthcare systems in the IoMT framework, focusing on the emerging features of Healthcare 5.0 with a patient-centric approach. It introduces a dedicated healthcare architecture addressing IoMT device security. We have also examined the services, applications, and potential research challenges of IoMT-based smart healthcare in a data-driven environment, which is of utmost importance. Future researchers in intelligent, connected, and customized healthcare systems are expected to concentrate on IoMT technological advancements and finding solutions to regulatory challenges, standardization challenges, infrastructure costs, existing network strain, security concerns, and privacy concerns within the healthcare system. Healthcare

system providers and patients will benefit from IoMT as researchers and practitioners offer solutions to these challenges. To address vulnerabilities and ensure security in intelligent healthcare systems, AI (DL or ML) algorithms are being applied to instigate a paradigm shift in data analytics within a data-driven environment. The implementation of a 5G/6G network with IoT next-generation technologies would enhance internet connectivity and alleviate network strain.

**Author Contributions:** Formal analysis, G.S.; Writing—review & editing, P.M. All authors have read and agreed to the published version of the manuscript.

**Funding:** This research received no external funding.

**Institutional Review Board Statement:** Not applicable.

**Informed Consent Statement:** Not applicable.

**Data Availability Statement:** Not applicable.

**Conflicts of Interest:** The authors declare no conflict of interest.

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
