# Peer review of "Internet of Medical Things Healthcare for Sustainable Smart Cities: Current Status and Future Prospects"

_applsci, doi:10.3390/app13158869_

Round 1

Reviewer 1 Report

      The work does not comply with the journal format; it must be rewritten. It would be best to reduce the keywords; they are too many and must be short. The abstract must be rewritten and adhere to the format guidelines. It would be best to improve your English; it is difficult to understand the text. Line 59 IoTs?? It is IoT, and IoMT is not declared, use the abbreviations well. The grammar of the work is poor The format of the tables is not correct; the note of Table 1 is others. In Figure 4(A) and (B), we must improve their quality and eliminate the graphs of circles. Lines 457 to 493 it does not correspond to the format. improve figures No section of previous works allows the proposal to be measured. There is no discussion about the expected results. The paper describes the concept of intelligent and connected healthcare, referring to the use of advanced technologies such as the Internet of Medical Things (IoMT), big data, cloud computing, artificial intelligence, and blockchain to provide more efficient healthcare services. , convenient and personalized. The article proposes to discuss the transformative potential of intelligent healthcare and the importance of collaboration between healthcare service providers, technology companies, and regulatory entities to address the associated challenges and ensure the privacy and security of patient data. In terms of approach, the article proposes a comprehensive review of existing literature, bibliometric data, and global marketing analysis to understand better these technologies and their impact on the healthcare system. However, in practice, the proposal was not executed, and there is no clear development of a method that allows for a literature review or proposes an approach where IoMT applications and services are explored.

English must be corrected.

Author Response

First and foremost, we would like to express our gratitude to the reviewer for bringing this aspect to our attention and helping us improve the manuscript. Below are the responses to the comments provided:

R1: The work does not comply with the journal format; it must be rewritten. 

The Author’s Response: In the revised manuscript, the authors have organized the paper according to the journal format.

R2: It would be best to reduce the keywords; they are too many and must be short. 

The Author’s Response: In the revised manuscript, the authors have reduced and used the relevant keywords as highlighted on page 1.

R3: The abstract must be rewritten and adhere to the format guidelines. It would be best to improve your English; it is difficult to understand the text. 

The Author’s Response: In the revised manuscript, the authors have rewritten the abstract as highlighted on page 1 and have followed the format guidelines with improved English.

R3: Line 59 IoTs?? It is IoT, and IoMT is not declared, use the abbreviations well. 

The Author’s Response: In the revised manuscript, the authors have corrected the typo as “IoT”.

R4: The grammar of the work is poor 

The Author’s Response: In the revised manuscript, the authors have made corrections to the grammar.

R5: The format of the tables is not correct; the note of Table 1 is others. 

The Author’s Response: In the revised manuscript, the authors have mentioned the caption of table 1, highlighted on page 2 under section 1.1.

R6: In Figure 4(A) and (B), we must improve their quality and eliminate the graphs of circles. 

The Author’s Response: In the revised manuscript, the authors have incorporated high-quality images and eliminated the graphs with circular shapes.

R7: Lines 457 to 493 it does not correspond to the format. improve figures No section of previous works allows the proposal to be measured. There is no discussion about the expected results. 

The Author’s Response: In the revised manuscript, the authors have followed the format guidelines for lines 457-493 which is now 391-432 under section 5.3 on page 13.

R8:The paper describes the concept of intelligent and connected healthcare, referring to the use of advanced technologies such as the Internet of Medical Things (IoMT), big data, cloud computing, artificial intelligence, and blockchain to provide more efficient healthcare services. , convenient and personalized. The article proposes to discuss the transformative potential of intelligent healthcare and the importance of collaboration between healthcare service providers, technology companies, and regulatory entities to address the associated challenges and ensure the privacy and security of patient data. In terms of approach, the article proposes a comprehensive review of existing literature, bibliometric data, and global marketing analysis to understand better these technologies and their impact on the healthcare system. However, in practice, the proposal was not executed, and there is no clear development of a method that allows for a literature review or proposes an approach where IoMT applications and services are explored.

The Author’s Response: In this paper, authors have extensively examined the concept of intelligent, interconnected, and customized healthcare systems within the framework of the Internet of Medical Things (IoMT). We have highlighted the integration of advanced technologies such as the IoMT, big data, cloud computing, artificial intelligence, and blockchain to enhance the efficiency, convenience, and personalization of healthcare services. To support our exploration, we have conducted a comprehensive review of existing literature, analyzed bibliometric data, and examined global market trends to gain valuable insights into these technologies and their impact on the healthcare landscape.

Moreover, our research has focused on discussing the key features and advancements of the Healthcare 5.0 paradigm, which represents the next evolutionary step in healthcare systems with a strong emphasis on personalized and patient-centric care. We have also showed a dedicated healthcare architecture tailored specifically for the IoMT, prioritizing the security considerations associated with medical devices and patient data. Lastly, we have dedicated attention to addressing critical research challenges, particularly those related to fundamental social needs. Ensuring equitable access to smart and connected healthcare systems has been a key focus, aiming to bridge the gaps in healthcare disparities and promote inclusivity.

By covering these aspects, we aim to contribute to the existing knowledge base and provide valuable insights into the potential of intelligent healthcare systems empowered by the IoMT and related technologies.

Reviewer 2 Report

The paper presents an overview of the Internet of Medical Things in the context of sustainable smart cities. It describes its enabling technologies, applications, challenges, and its development in the near future.

Both the topic and the paper should result of interest to the readers, as it deals with technology that is being developed and deployed worldwide to solve real problems. However, authors should review the following concerns:

- The paper seems a bit repetitive. It is perhaps a matter of how it is organized but also some corrections to the writing style are needed. Check the text in lines 66-72; it seems like repeating the same idea three times. This same style is found several times through the whole document.

- My first real concern comes with section 3. There is no description on how the bibliometric analysis was conducted or what were the goals of the study (though this might be inferred from the text). In this same section, text in lines 255-257 appears to be from the paper template.

- The timeline in Fig. 5 is confusing, being ticked by centuries. In particular the location of Healthcare 2.0 doesn't match a period where the mentioned technologies were available. This whole section goes almost without references.

- Please describe in more depth the differences between Healthcare 4.0 and 5.0. Sections 4.4 and 4.5 are very similar in content, and the descriptions in figure 5 are identical.

- At the end of the first paragraph in section 5 there is a mention to something that is expected to happen by 2019. Please be more careful to how this references are presented. Did it happen or no? It can be described as how it was envisioned and what actually happened.

- The second paragraph in that same section needs to ve reviewed and probably rewritten. 

- Another of the concerns comes with the architecture shown in Fig. 7. In particular with the description of the Physical and Data layers, that are shown and described together, and their relationship with the gateway layer. The description of the Gateway layer seems correct: "ensures that the data is transmitted securely and efficiently between the devices and the data layer". However, this is not what is shown in the figure, as the gateway layer connects to the application layer. In a typical IoT architecture, the data layer would be located in the cloud, where data is stored, processed, and access to these data is provided to applications and users. Sensors (physical layer) are the ones to obtain data, but they relay data to the cloud or some local storage mechanism.

- Section 7 focuses on the applications. This section lacks references for most of the application areas described. In addition, applications and areas are repeated in some sections. For instance, robotics are described in two sections, one as general application and the other as a specific one. The same applies to telemedicine, medication management/drug delivery, and chronical diseases that could be put together in the same section.

There are several grammar issues across the document. Nothing that impedes understanding what authors are stating, but noticeable.

Author Response

First and foremost, we would like to express our gratitude to the reviewer for bringing this aspect to our attention and helping us improve the manuscript. Below are the responses to the comments provided.

R1: The paper seems a bit repetitive. It is perhaps a matter of how it is organized but also some corrections to the writing style are needed. Check the text in lines 66-72; it seems like repeating the same idea three times. This same style is found several times through the whole document.

The Author’s Response: In the revised manuscript, the authors have eliminated the repetitive pattern. Also revised the introduction and whole document to avoid the repetition of same idea.

R2: My first real concern comes with section 3. There is no description on how the bibliometric analysis was conducted or what the goals of the study were (though this might be inferred from the text). In this same section, text in lines 255-257 appears to be from the paper template.

The Author’s Response: In the revised manuscript, the authors have added the bibliometric analysis of IoMT healthcare in smart cities, using data from the Scopus database, focused on articles as the primary publication type. Through visualization analysis using VOS viewer software, the study identified research gaps in the IoMT for smart cities field. This analysis offers valuable insights for researchers looking to explore unexplored areas within the domain, aiding in the identification of research trends and areas for future investigation. This is highlighted under Section 3 on page no. 7 and 8.

R3: The timeline in Fig. 5 is confusing, being ticked by centuries. In particular the location of Healthcare 2.0 doesn't match a period where the mentioned technologies were available. This whole section goes almost without references.

The Author’s Response: In the revised manuscript, the authors have modified the Fig. 5 on page no. 8. Also authors have added references [38],[39] and [40] related to the field.

  1. Li, J.; Carayon, P. Health Care 4.0: A vision for smart and connected health care. IISE Transactions on Healthcare Systems Engineering 202111, 171-180.
  2. Aceto, G.; Persico, V.; Pescape, A. Industry 4.0 and health: Internet of things, big data, and cloud computing for healthcare 4.0. Journal of Industrial Information Integration 202018, 1-13.
  3. Mbunge, E.; Muchemwa, B.; Batani, J. Sensors and healthcare 5.0: transformative shift in virtual care through emerging digital health technologies. Global Health Journal 2021, 5, 169-77.

R4: Please describe in more depth the differences between Healthcare 4.0 and 5.0. Sections 4.4 and 4.5 are very similar in content, and the descriptions in figure 5 are identical.

The Author’s Response: In the revised manuscript, the authors have modified the description in Fig. 5, and also explained the healthcare 4.0 and 5.0. Its shows differences now because Healthcare 4.0 and Healthcare 5.0 are two distinct paradigms of healthcare that represent different stages of transformation driven by advancements in technology and patient-centered care. Healthcare 4.0 primarily focuses on the integration and utilization of advanced technologies such as artificial intelligence (AI), Internet of Things (IoT), big data analytics, and robotics in healthcare systems. On the other hand, Healthcare 5.0 takes a more human-centered approach, placing a strong emphasis on holistic well-being and patient engagement. The content is highlighted under section 4, page no. 8 and 9.

R5: At the end of the first paragraph in section 5 there is a mention to something that is expected to happen by 2019. Please be more careful to how this references are presented. Did it happen or no? It can be described as how it was envisioned and what actually happened.

The Author’s Response: In the revised manuscript, the authors have eliminated this point.

R6: The second paragraph in that same section needs to ve reviewed and probably rewritten. 

The Author’s Response: In the revised manuscript, the authors have reviewed and rewritten the paragraph.

R7: Another of the concerns comes with the architecture shown in Fig. 7. In particular with the description of the Physical and Data layers, that are shown and described together, and their relationship with the gateway layer. The description of the Gateway layer seems correct: "ensures that the data is transmitted securely and efficiently between the devices and the data layer". However, this is not what is shown in the figure, as the gateway layer connects to the application layer. In a typical IoT architecture, the data layer would be located in the cloud, where data is stored, processed, and access to these data is provided to applications and users. Sensors (physical layer) are the ones to obtain data, but they relay data to the cloud or some local storage mechanism.

The Author’s Response: In the revised manuscript, the authors have modified the Fig. 7 and the content, which is highlighted under section 5.3 on page no.12 and 13.

R8: Section 7 focuses on the applications. This section lacks references for most of the application areas described. In addition, applications and areas are repeated in some sections. For instance, robotics are described in two sections, one as general application and the other as a specific one. The same applies to telemedicine, medication management/drug delivery, and chronical diseases that could be put together in the same section.

The Author’s Response: In the revised manuscript, the authors have modified this section and merged repeated applications. Now in all there are 16 points.

Round 2

Reviewer 2 Report

Authors have attended previous comments and concerns in a proper manner. The review is interesting and covers several aspects of the internet of medical things.

However, this revised version still has a single concern. The abstract implies that the architecture is in some way developed by the authors ("We introduce..."), something that later in the document makes suppose it is not, as there is a reference to a work from other authors. Additionally, mentioning the architecture as one of the highlights in the abstract makes you expect that it will be also highlighted in other sections, but it is only found when it is described, and not even the conclusions include something about the architecture.

Grammar and style were really improved. Please check for minor edits.

Author Response

First and foremost, we would like to express our gratitude to the reviewer for bringing this aspect to our attention and helping us improve the manuscript. Below is the response to the comments provided.

R1: Authors have attended previous comments and concerns in a proper manner. The review is interesting and covers several aspects of the internet of medical things.

However, this revised version still has a single concern. The abstract implies that the architecture is in some way developed by the authors ("We introduce..."), something that later in the document makes suppose it is not, as there is a reference to a work from other authors. Additionally, mentioning the architecture as one of the highlights in the abstract makes you expect that it will be also highlighted in other sections, but it is only found when it is described, and not even the conclusions include something about the architecture.

The Author’s Response:

In the revised manuscript, the authors have implemented the reviewer’s comments and suggestions. It is mentioned in the abstract ("We introduce...") as highlighted on page no. 1, is developed by the authors. We have removed the reference no. [42] from section 5.3 on page no. 12, as it was creating confusion that the architecture is taken from this reference paper.  

We have also mentioned about the architecture in conclusion as highlighted in Section 9 on page no. 25, has also given below.

(“This paper explores intelligent, personalized healthcare systems in the IoMT framework, focusing on the emerging features of Healthcare 5.0 with a patient-centric approach. It introduces a dedicated healthcare architecture addressing IoMT device security.”)